# Exploiting the signal to noise ratio in multi-system predictions of boreal summer precipitation and temperature

**Juan C. Acosta Navarro[1] and Andrea Toreti[1]**

[1]European Commission, Joint Research Centre, Ispra, Italy.

*Correspondence to:* Juan C Acosta Navarro (juan.acosta-navarro@ec.europa.eu)

**Abstract**

Droughts and heatwaves are among the most impactful climate extremes. Their co-occurrence can have adverse consequences on natural and human systems. Early information on their possible occurrence on seasonal timescales is beneficial for many stakeholders. Seasonal climate forecasts have become openly available to the community but a wider use is currently hindered by limited skill in certain regions and seasons. Here we show that a simple forecast metric from a multi-system ensemble, the signal to noise ratio, can help overcome some limitations. Forecasts of mean daily near surface air temperature and precipitation in boreal summers with high signal to noise ratio tend to coincide with observed larger deviations from the mean than summers with small signal to noise ratio. The signal to noise ratio of the ensemble predictions may serve as a complementary measure of forecast reliability that could benefit users of climate predictions.

## 1. Introduction

Droughts are typically slow onset climate extreme events (Mishra and Singh, 2010), yet they can be disruptive and affect millions of people every year (Below et al., 2007; Enekel et al., 2020). Heatwaves can intensify and trigger a faster drought evolution (Bevacqua et al., 2022). Compound drought and heatwaves can strongly impact socio-economic and ecological systems, and may even compromise our ability to reach the UN sustainable development goal on climate action while strongly reducing the Earth system's current natural capacity to absorb and store carbon (Yin et al., 2023). The use of seasonal climate forecasts can provide actionable information to reduce the risks and the impacts of these events on key sectors like agriculture, energy, transport, water supply (Buontempo et al 2018; Ceglar and Toreti 2021).

In the last couple of decades, climate predictions have shown important progress in anticipating the evolution of various components of the climate system across the subseasonal to decadal time range (Merryfield et al., 2020; Meehl et al., 2021). A combination of multiple forecast systems has shown overall benefits as compared with single systems, and can improve forecast quality up to a certain extent (Hagedorn et al., 2005; Mishra et al., 2019). In spite of the recent progress, climate predictions still exhibit low to moderate skill in many regions and seasons (e.g. European summer; Mishra et al. 2019), something that limits their use and represents a barrier for stakeholders. Furthermore, multiple studies have shown that large ensembles are required to achieve skillful predictions, something that seems to be related to the forecast systems being more skillful at predicting real climate

than at predicting their own realizations (i.e. ensemble members). This odd phenomenon has been called the signal
to noise paradox (Eade et al., 2014; Scaife and Smith, 2018; Smith et al., 2020). It is particularly evident in the
Euro Atlantic region during winter both on seasonal and decadal timescales. However boreal summer predictions
have been generally overlooked. A recent study based on a single forecasting system has shown that sampling
years with high SNR results in more skillful predictions of monthly temperatures in Japan throughout the year
(Doi et al., 2022).

In this study we exploit multi-system ensembles to test whether specific boreal summers with higher than normal
predictability can be detected through the local relation between skill and SNR. We explore this for near surface
air temperature and precipitation, both locally and on large aggregated mid-latitude regions of the Northern
Hemisphere.

**2. Methods**

This analysis is based on seasonal re-forecasts (also known as hindcasts) of mean boreal summer precipitation
and 2-meter mean daily temperature (T2m) for the period 1993-2016 from ECMWF SEAS5 (S5, Johnson et al.,
2019), UKMO GloSea6 (S600, MacLachlan et al., 2015), MeteoFrance (S8, Batté et al., 2021), CMCC (S35,
Gualdi et al., 2020) and DWD (S21, Baehr et al., 2015), available from the Copernicus C3S Climate Data Store.
The observationally based datasets to evaluate the re-forecasts are ERA5 (Hersbach al., 2020) for T2m and GPCC
(Schnider et al., 2011) for precipitation. The use of summer mean T2m is not intended to characterize single
heatwaves, but to estimate average daily deviations from the mean on a seasonal scale. In a climatological sense,
more intense, more frequent or longer heatwaves than usual generally define hot summers and hence average T2m
may be seen as a seasonal integrator of heatwave activity. Forecast skill is evaluated with the anomaly correlation
coefficient (ACC) between the ensemble mean and the observational reference. To complement the skill estimates
of ACC, two additional deterministic skill metrics are computed: the mean squared skill score (MSSS, Murphy,
1988) and the Gilbert skill score (GSS, WMO, 2014). The mean squared skill score compares the mean square
error of the forecasts with the mean square error of the climatological value. It ranges from minus infinity to 1
and values above 0 indicate skill in the predictions. The GSS measures the fraction of correctly predicted events
over the total number of predicted events plus misses, and takes into consideration the randomly predicted events.
The thresholds to define event/non event are the top and bottom 25% summers for T2m (hot) and precipitation
(dry), respectively. Standardization of the anomalies of each ensemble member and the observational reference
data is performed prior to the analysis. This step guarantees that each member from each system has a comparable
year-to-year variability to the observed one. Additionally, the standardized T2m anomalies are linearly detrended
at the grid level and for each member of the re-forecasts and in ERA5 to isolate as much as possible the impact of
the long term warming.

Following Doi et al. (2022), the SNR is calculated as: $SNR = \frac{\mu_e}{\sigma_e}$, where $\mu_e$ is the multi-system ensemble mean
and $\sigma_e$ is the multi-system standard deviation after standardization, computed across ensemble members for every
summer (June - August) and for each gridbox. 25 members per system are used to have an equal contribution from
each system.

**3. Signal to noise ratio and forecast skill**


Figure 1 displays spatial maps of mean (boreal) summer T2m ACC, time averaged SNR, and a scatter plot which shows the local relation between ACC and SNR. On average, skill values over land increase with higher SNR values. Negative values of ACC are nearly non-existent when the threshold of SNR exceeds the value of about 0.5 in the same gridbox. Statistically significant skill in T2m is mostly confined to the tropics and sub-tropics. However, significant skill is also found in western North America, the eastern Mediterranean, central Asia and southern South America. Notable exceptions in the tropics are Congo and parts of the Amazon rainforests. The patterns of SNR largely mirror those of ACC. Generally, there is a good agreement between areas of high skill (ACC) and areas with high SNR, something that is further confirmed by the local relation between ACC and SNR (Fig. 1c).

91

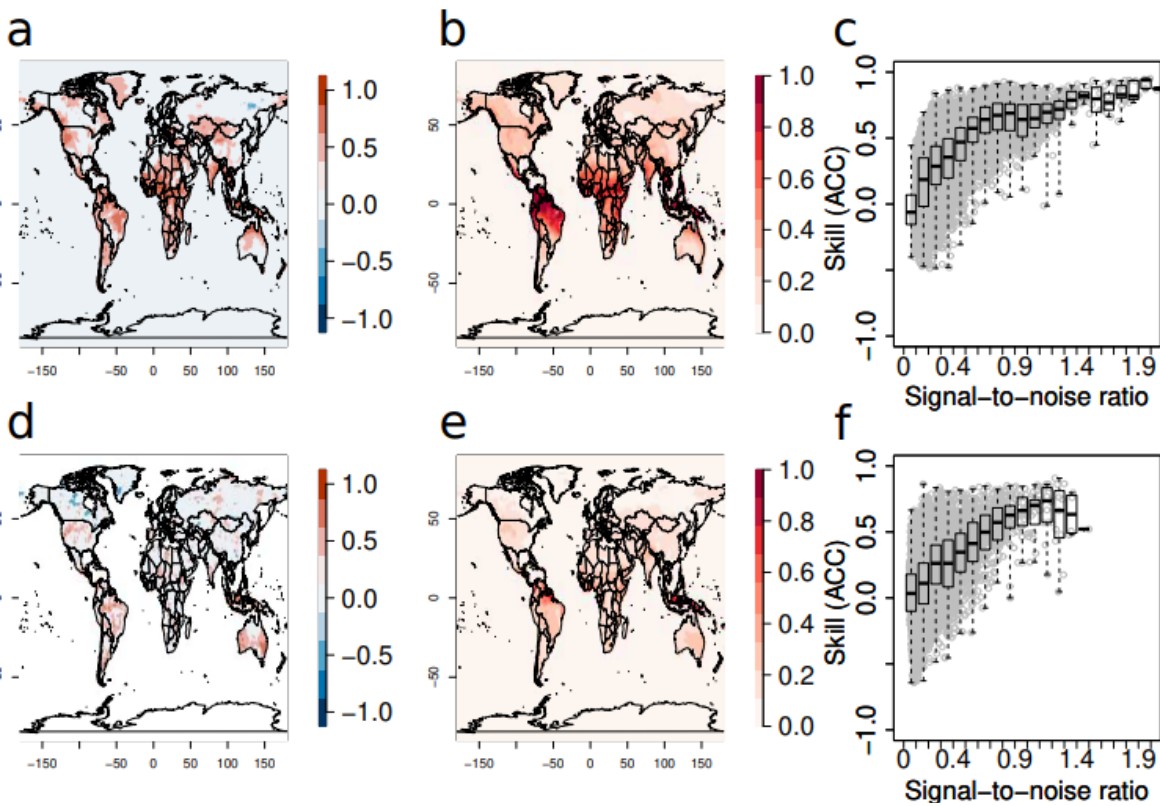

92

**Figure 1: June-August Skill (ACC), time averaged SNR and scatterplots of local relation between ACC and SNR for T2m (a-c) and precipitation (d-f). Each gray dot in (c,f) represents the values of ACC and SNR at each gridbox. Only statistically significant values with a 90% confidence based on a t-test are displayed in (a,d). The re-forecasts are initialized every May.**

97

Precipitation follows a similar behavior in terms of ACC and SNR, although statistically significant skill is less widespread (Fig. 1d-f). Areas under the influence of El Niño Southern Oscillation (ENSO; Lenssen et al., 2020) appear as regions with significant ACC and high SNR. Skillful values are mostly located in the Americas, the Maritime continent and Australia. Precipitation skill and SNR in Africa and Asia are much lower, making these the regions with the largest qualitative differences between the two variables.

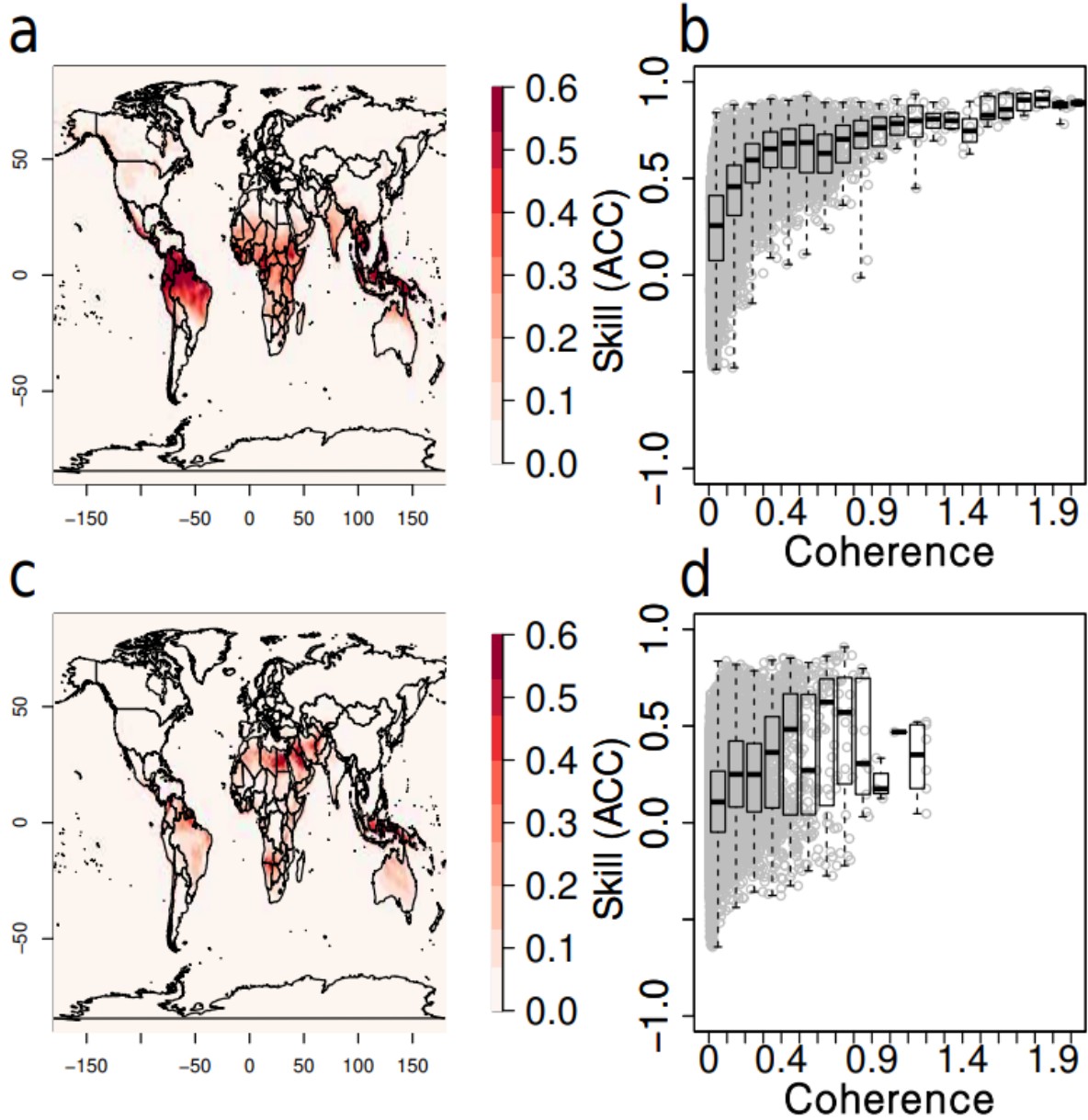

**Figure 2: June-August time averaged ensemble coherence and scatterplots of the local relation between ACC and ensemble coherence for T2m (a-b) and precipitation (c-d). Each gray dot in (b,d) represents the values of ACC and ensemble coherence at each gridbox. The re-forecasts are initialized every May.**

In Figure 2 we show the effect of the ensemble coherence on skill. Ensemble coherence is defined as the inverse of the ensemble standard deviation ($\sigma_e$), minus one. The spatial distribution of time averaged ensemble coherence displays many similarities to the SNR for both T2m and precipitation, although the signal is clearly dominated by the tropics and subtropics with virtually no contribution from the extra-tropics, except for a minor one from T2m in western North America and from precipitation in the Middle East (Fig. 2a,c). In terms of the local relation between ensemble coherence and skill, T2m displays a clear increase in skill with higher values of coherence (Fig. 2b). Skill is virtually always positive when coherence values exceed 0.3, implying that ensemble spread may also be a good indicator of skill for T2m, similar to SNR. For precipitation there is weaker relation between skill and

ensemble coherence than for T2m as there appear to be as many locations of high coherence with little skill as
locations with high skill and high coherence (Fig. 2d). This can be a result of a weaker relation between skill and
ensemble coherence than between skill and SNR, but may also be at least partially a result of the large uncertainty
in observed precipitation in many regions.

Based on the observed link between skill and SNR, we use the latter one as the single criterion to exclude from
the re-forecasts years with very low and very high values to understand their impact on skill. When 25% of the
years (6 in total) with the highest SNR (Fig. 3a) are excluded, the results overall show much lower values of ACC
than when only 25% of the years with the lowest SNR are excluded (Fig. 3b). Furthermore, differences between
the latter and the former result (in many cases) in higher statistically significant values than the ACC computed
when selecting only years without the highest SNR (Fig. 3a,c). This result highlights the importance that these
extreme SNR years can have on skill. In fact, only skill values that are computed by excluding the bottom 25% of
SNR years (Fig. 3b) are comparable to the ones estimated when all years are used for the computation (Fig. 1a).

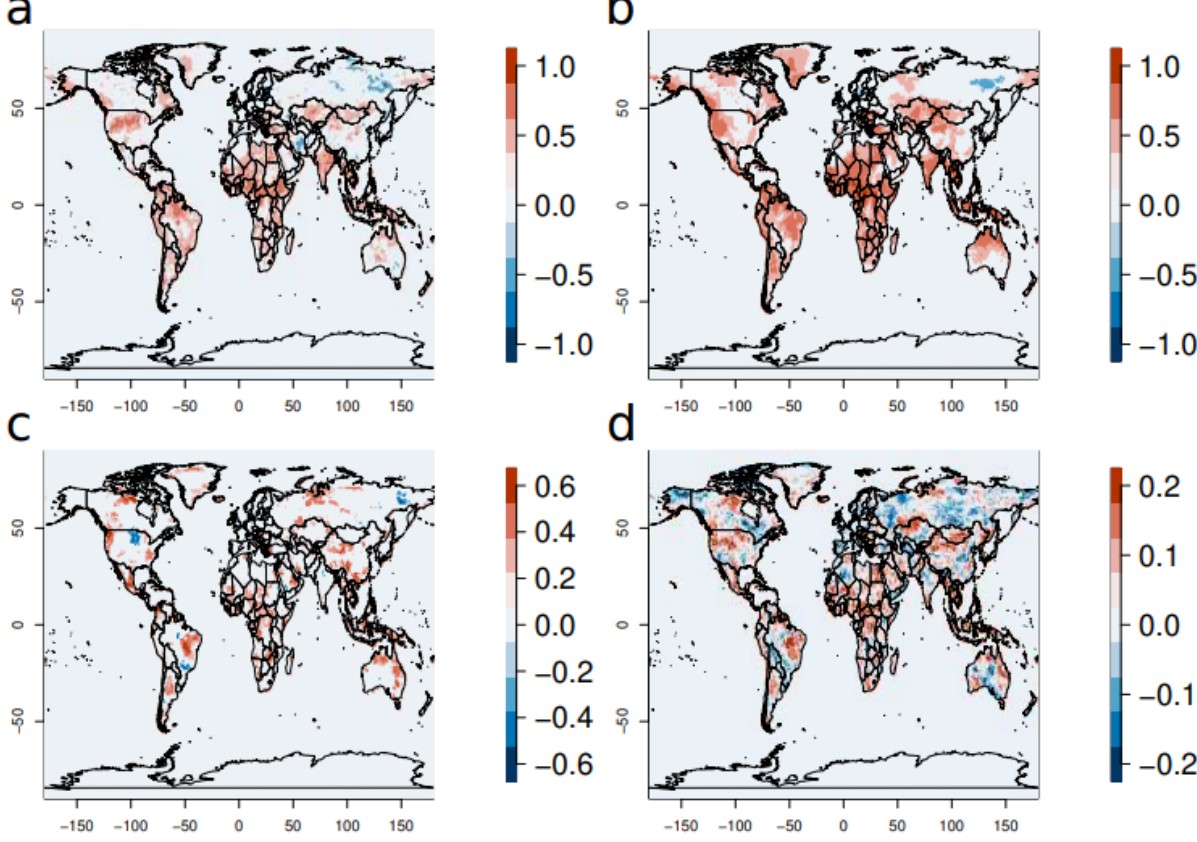

**Figure 3: Skill (ACC) of T2m predictions excluding 25% of the years with highest (a) and lowest (b) local SNR. (c)**
**Difference between (a) and (b). (d) Difference in the time-averaged absolute deviation from the mean in ERA5 T2m,**
**excluding years having 25% of the lowest and highest local SNR, respectively. Only statistically significant values with**
**a 90% confidence based on a t-test are displayed in (a-c). The re-forecasts are initialized every May.**

Interestingly, using the same criterion to select ERA5 T2m values reveals that in general, excluding years with
high ensemble SNR results in lower absolute deviations from the mean than when the low SNR years are excluded
(Fig. 3d). Additionally, these differences overall coincide with regions with significant skill differences (Fig.
3c,d). This implies that years with more extreme deviations from the mean (in the observations/reanalysis) may
be identified a priori by calculating the ensemble SNR of the forecast, and that forecast systems are in general
more skillful when large deviations from the mean occur.

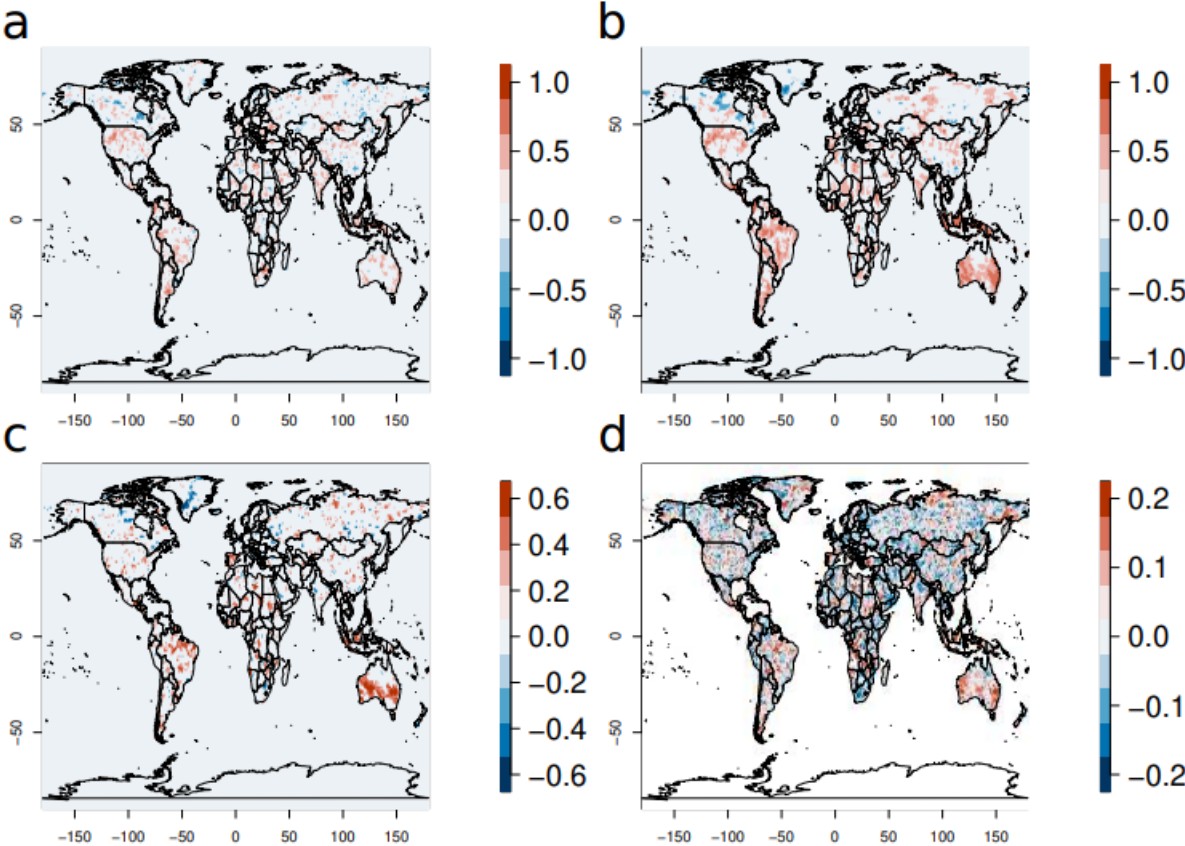

**Figure 4: Same as Figure 3, but for precipitation.**
Similar to T2m, the exclusion of years with high SNR also results in lower overall precipitation skill values than
the one obtained when excluding low SNR years (Fig. 4a,b). Important skill differences appear in the Iberian
Peninsula, Brazil, Australia and Indonesia (Fig. 4c), and in most cases imply a shift from non-significant to
significant skill (Fig. 4 a and b, respectively). Contrasting with T2m, the relation between ACC and mean absolute
deviation from the mean in the observations is not obvious for precipitation (Fig. 4c,d). To further investigate this
behavior, we analyzed the relationship between skill differences and the differences in absolute deviation from
the mean for T2m and precipitation, as usual by using the re-forecasts that exclude the 25% of the years with the
lowest and the highest SNR, respectively. This analysis (not shown) confirms a statistically robust relationship
between skill and large deviations from mean observed precipitation, but still weaker than for T2m.

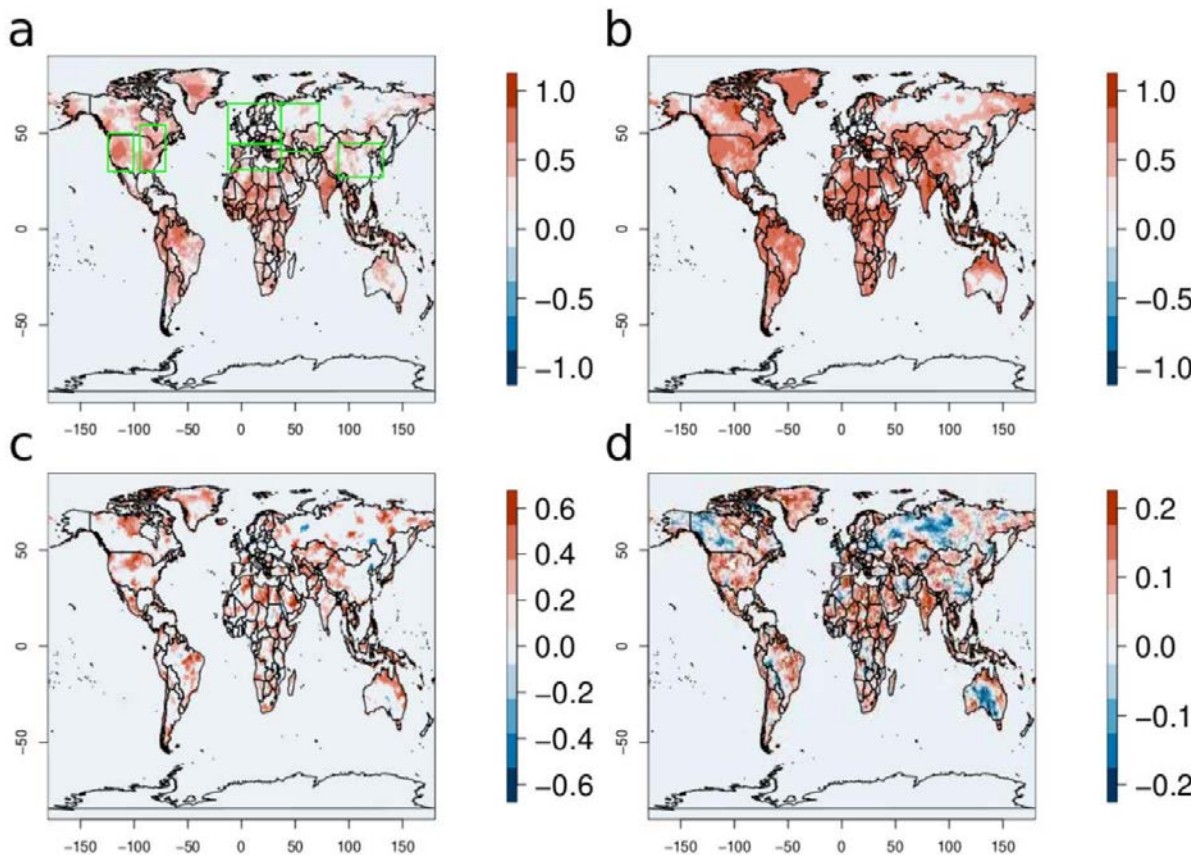

**Figure 5: The same as Figure 3, but for re-forecasts initialized every June. Boxes in (a) show the areas used in Figures 6 and 7.**

Figure 5 shows a clearer relation between the impact on skill of the most extreme years in terms of SNR and the absolute T2m anomalies in ERA5, as compared with Figure 3. There is a good correspondence in all continents, including parts of Europe (Fig. 5 c,d). The only difference between the two Figures is that they show the results from re-forecasts with different initialization dates. Both target the boreal summer months (June-August), but Figure 3 shows the results from the May initialization while Figure 5 shows the results from the June initialization. Similar qualitative conclusions can be made for precipitation (not shown).

In Figure 6 we use the same methodology to sample years based on T2m SNR, but applied to specific northern hemisphere mid-latitude regions: the Mediterranean, North and Central Europe, north western Asia, east Asia, western North America and eastern North America. All the three skill metrics computed show that sampling the 18 years with highest SNR, generally results in more skillful T2m predictions than when sampling all 24 years or the 18 years with lowest SNR. The only exceptions are observed in North and Central Europe, where there is basically no skill, as well as in eastern North America, where all the three selection methods show similar skill levels. Examples of successful prediction of extreme (high) T2m years and high SNR are 1999 and 2003 in the Mediterranean, 2002 in northern/central Europe, 1998 in northwestern Asia, 2006 and 1998 in western and eastern North America, respectively. There are also some examples of extreme (high) T2m and low SNR, such as 2012 in the Mediterranean, or 1994 and 2016 in East Asia. However, higher overall GSS for the top T2m positive anomalies indicates that on average, sampling years with high SNR results in better prediction of the extreme events.


A similar analysis on precipitation is shown in Figure 7. The results of precipitation qualitatively agree with those
of T2m. Precipitation skill is highest for years with highest SNR and lowest for years with lowest SNR, the only
exception being northern/central Europe, again a region with no skill in either precipitation or T2m predictions.
Years of successful predictions of low precipitation and high SNR are 1994 and 2000 in the Mediterranean, 2015
in northern/central Europe, 1997 and 2001 in East Asia, 2003 in western North America, and 2011 in eastern
North America. Similar to T2m, GSS for low precipitation summers is generally higher for the top 18 years (in
terms of SNR) than for the bottom 18 years or for all 24 years. It is worth noting that skill scores for precipitation
are generally lower than those of T2m. This is primarily due to the lower overall predictability of precipitation
compared to T2m.  . Note also that the same conclusions are obtained for both T2m and precipitation when
separately sampling only the half of years with highest and lowest SNRs and/or when varying the threshold to
define the most extreme years used in the GSS calculations (not shown).

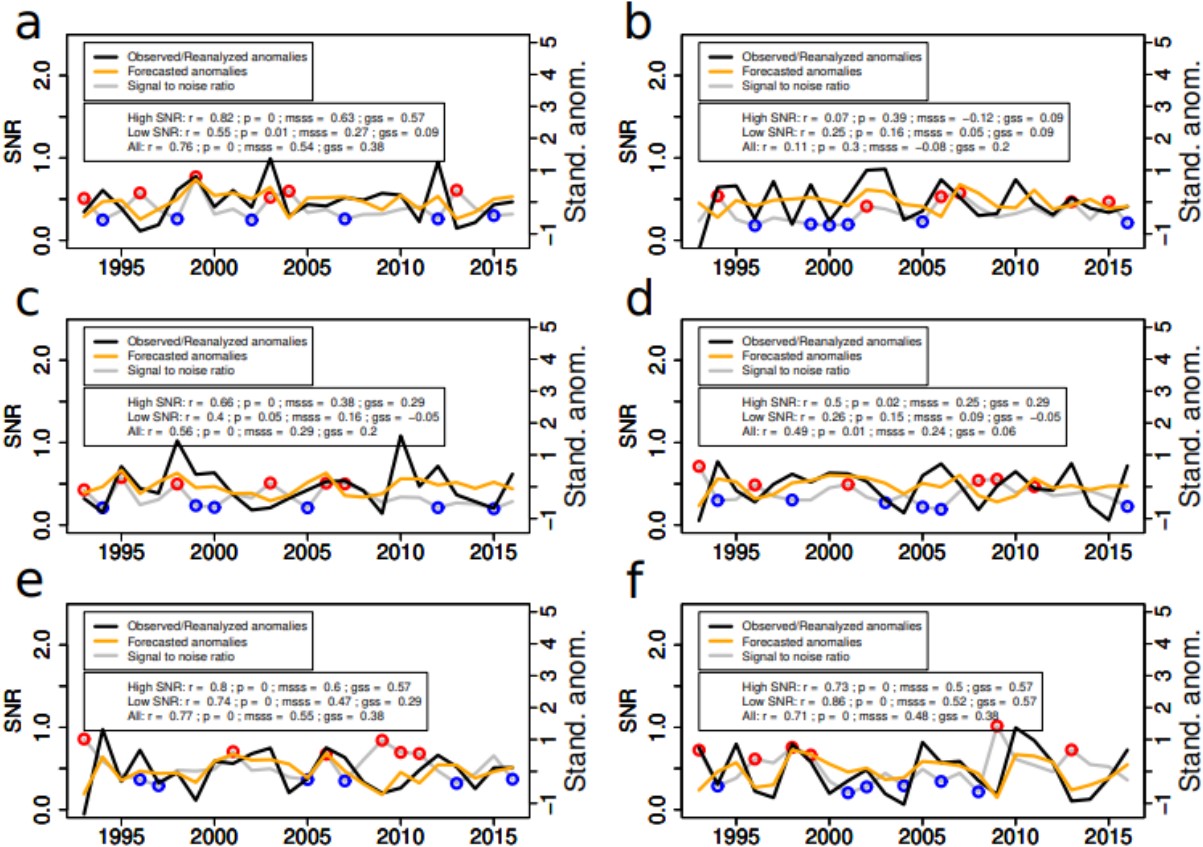


**Figure 6: Area-averaged time series of observed and predicted, detrended and standardized mean summer T2m (right**
**axis) and SNR (left axis) in (a) the Mediterranean (10W-35E, 30-45N), (b) North and Central Europe (10W-35E, 45-**
**65N), (c) northwestern Asia (35-70E, 40-65N), (d) East Asia (90-130E, 25-45N), (e) western North America  (123-100W,**
**30-50N) and (f) eastern North America (90-70W, 30-55N). Skill metrics are provided separately for the 18 years with**
**highest SNR (excluding blue circles), the 18 years with the lowest SNR (excluding red circles) and for all 24 years. The**
**skill metrics are linear correlation, mean square skill score and Gilbert skill score (See methods). The p-values of the**
**linear correlation coefficients are also displayed for each region. The results are from the re-forecasts initialized in**
**June.**

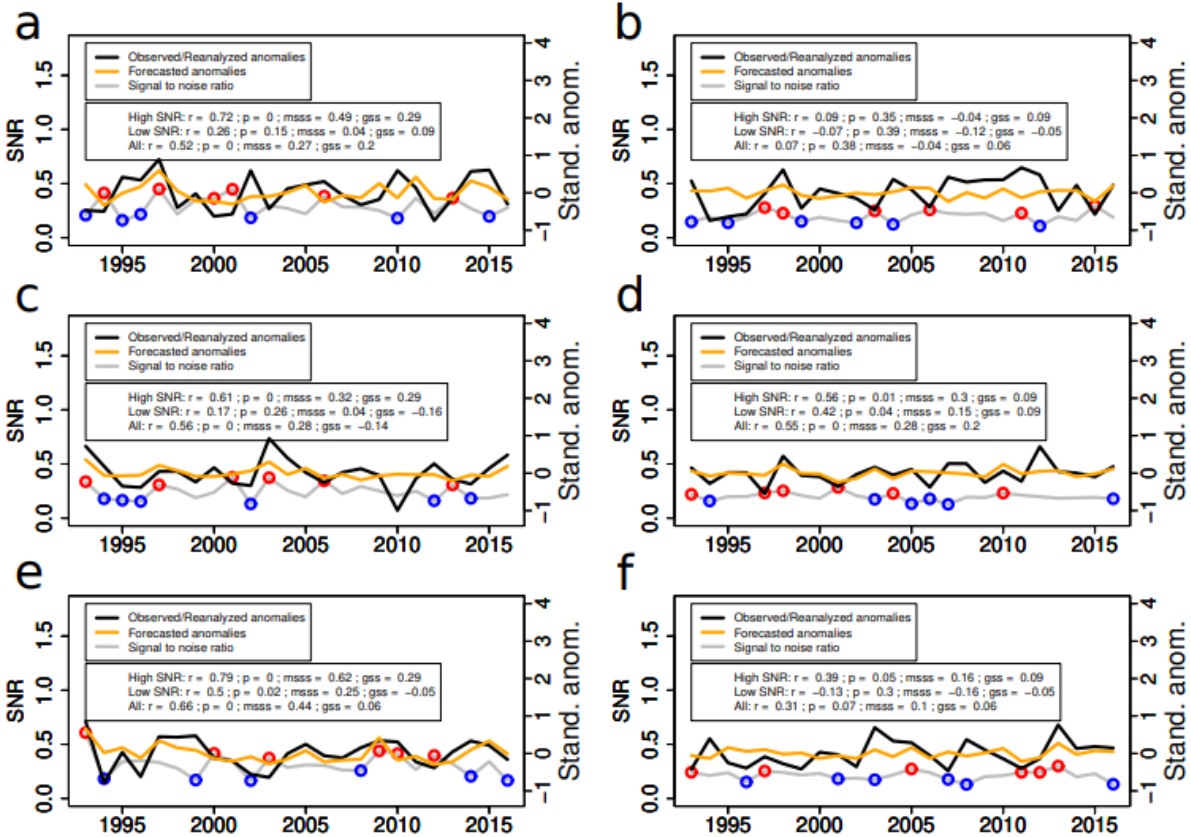

Figure 7: The same as Figure 6, but for precipitation.

**4. Discussion**

The SNR measures the relative weight of the ensemble mean anomalies with respect to the ensemble coherence. Its close resemblance in terms of spatial patterns with a skill metric like ACC indicates that it can provide complementary information related to seasonal climate predictability. We have shown that in regions where the forecasts are skillful, years with high SNR exhibit on average larger observed deviations from the mean than years with low SNR, for both T2m and precipitation. This means that forecast systems are on average more reliable at predicting extremes when there is a higher coherence. This has been further demonstrated for several Northern Hemisphere mid-latitude regions during boreal summer. Ensemble coherence is also a good indicator of T2m and precipitation predictability, although appears to be only suitable for tropical and subtropical locations.

Despite the well-known limitations of climate forecast systems (e.g. the signal to noise paradox), we have shown that in a multi-system ensemble, the SNR may provide valuable information as it represents an intrinsic measure of reliability for T2m and precipitation forecast. The short span of 24 years defining the common hindcast period is a limitation of this study. Hence, longer hindcasts would be necessary to obtain more robust results, but are currently unavailable for most of the multiple systems analyzed.

**Data availability**

All the data used in this publication is open. The R scripts used in the analysis are available upon request.

225

**Author contribution**

JCAN and AT conceptualized the study and developed the methodology. JCAN performed all analyses and wrote

the first draft. Both co-authors contributed with the preparation of the final draft.

**Acknowledgements**

The authors would like to acknowledge two anonymous reviewers who provided valuable comments and helped

improve the final version of this study.

**Competing interests**

The contact author has declared that none of the authors has any competing interests.

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
