# Peer review of "Exploiting the signal to noise ratio in multi-system predictions"

_EGUsphere, 2023_

## Referee Comment (RC1)

**Review for "Exploiting the signal to noise ratio in multi-system predictions of summertime precipitation and maximum temperatures in Europe", Acosta-Navarro and Toreti**

**General comments**

This study reveals how the ratio between ensemble mean and ensemble spread in dynamical seasonal forecasts for boreal summer can inform in advance about potentially large temperature departures from climatology. This is a valuable asset, considering the limited predictability over Europe. The paper also indicates that this metric can help identify potential predictability drivers, with the example of a North Atlantic spring SST Atlantic preceding anomalously warm summers over Europe.

Overall the scientific question is simple and clear, and the results are quite convincing. Additionally, the paper is well written and enjoyable to read. This paper deserves to be published in WCD, after considering a few mostly minor revisions.

The introduction and discussion sections are particularly short. It is an asset for the introduction since it is sufficiently precise to properly state the scientific question addressed, and goes straight to the point. On the other hand, the results shown throughout the paper raise a number of questions that would deserve to be at least pinpointed in the discussion, if the authors want to keep the scope as it is. The main point to address concerns the section focusing on predictability source, and is detailed hereafter.
Finally some figures need to be be improved, as suggested below, to facilitate the reading and understanding.

**Specific comments**

L.42: "Tmax may be seen as a seasonal integrator of heatwave activity".
No doubt that your approach is valid, but wouldn't it have been simpler to use the seasonal average of daily mean temperature, rather than daily maximum temperatures ?
The departure of daily mean temperatures from climatology also permits to identify heatwaves, and it integrates the full diurnal cycle, which may be relevant since night-time temperature is also impactful during heatwaves.

L. 86-88 : This is not obvious from your figures. Considering this comment, we probably expect to see blue shades over NW Europe in fig 2.c but it is barely visible on your maps: the contrast between dark gray and blue is weak. I would suggest either to improve the maps (you may remove country borders, or pick a lighter gray shade for non significant values, or alternatively hatch/stipple where values are statistically significant.) or to provide additional support to your assertion.

If fig. 5d shows the mean 1994 and 2003 JJA, one might wonder if the SST dipole in the Atlantic is present equivalently in both summers, or prevails in one of the two summers. Since spring 2003 does not stand out as a season with particularly high spring SST index (fig. 5c), this question is even more intriguing. You could provide in appendix or supplementary material separate maps for both summers and comment on the specific case of 2003 (weak dipole in spring, stronger in summer ?)

L. 144-146: In my view, the subsampling of members based on the May index for re-forecast initialized in May makes no sense because the ensemble spread is presumably too small. I guess Mid-Atlantic SST anomalies are somewhat persistent, and it takes time for ensemble members to diverge from each other (and from SST initial conditions too). I think a discussion on the spread would be valuable here.

Overall, in section 3.2, the distinction between spring and summer SST patterns is a bit overlooked, and gives an impression of 'cherry picking'. Which one (spring or summer) is better correlated with warm summer anomalies over Europe ? In figure 5 you build your index on spring patterns, then you show summer patterns for 1994 and 2003, but the 2003 spring index does not stand out, and this point is not even commented on. Finally, in the subsampling procedure, you admit that it does not work when the subsampling is based on the May SST index. Besides the question of the relevance of subsampling too early after initialization (see point above), how could your findings help forecasters in real-time ? Would it be more advisable for them to systematically subsample their summer forecast ensemble, even if there is very little match with the SST pattern, or should they (also? instead?) evaluate a kind of distance metric or spatial correlation for each member to the SST dipole pattern, because maybe the SST pattern is not relevant every year.

References:

The reference of the MeteoFrance system is the following
Batté L., L. Dorel, C. Ardilouze, and J.-F. Guérémy, 2017: Documentation of the METEO-FRANCE seasonal forecasting system 8. Météo-France, 36 pp., https://www.umr-cnrm.fr/IMG/pdf/system8-technical.pdf.

L.38 Since you have retrieved the forecast data from the C3S Climate datastore, I think you should cite it and acknowledge properly, as recommended here:

https://confluence.ecmwf.int/display/CKB/
How+to+acknowledge+and+cite+a+Climate+Data+Store+%28CDS
%29+catalogue+entry+and+the+data+published+as+part+of+it

And here: https://cds.climate.copernicus.eu/cdsapp#!/dataset/seasonal-original-single-levels?tab=doc

**Technical corrections**

L.48 Typo: It should read "from ERSSTv5 (Huang et al., 2017) and ERA5, respectively."

Figure 1 : according to fig. 1c and 1f, a few grid points have SNR values exceeding 1. Thus, this should also show in the colorbar of fig 1b and 1e which looks bounded by 1

Figure 5 : in the caption, the reference to subplot (e) is missing

Figure 6:  When zooming in, I can see gray dots everywhere. This figure should be improved.
In addition, the color bars are different for temperature and precipitation. The same color bar would help compare the skill improvements for both variables. Finally, the color bars themselves are confusing: they should be improved by aligning the labels with color bounds.

L. 110-111: There is a mistake in the description of the ERA5 gray line in Fig5a/b, which gave me a hard time to understand: it is not the absolute value of the standardized ERA5 Tmax anomalies that you show, but the absolute deviation from mean! Luckily, the caption of figure 5 is okay.

L. 116: Missing specification that this statement refers to fig.5b

L. 117-119 : For these years with high ensemble SNR and high absolute anomalies in observations, I assume that the sign of the model re-forecast anomaly is the same as the sign of the observed anomaly. If so, this should be stated in some way, or overlaid in fig 5a/5b if feasible.

L. 142 : Fig 6a,c and 6b,d, not 5

L.145: typo 're-forecasts'

---

## Author Response (AR1)

Review 1

We thank the anonymous reviewer for his/her useful and constructive comments.

Review 1 for "Exploiting the signal to noise ratio in multi-system predictions of summertime precipitation and maximum temperatures in Europe", Acosta-Navarro and Toreti

General comments

This study reveals how the ratio between ensemble mean and ensemble spread in dynamical seasonal forecasts for boreal summer can inform in advance about potentially large temperature departures from climatology. This is a valuable asset, considering the limited predictability over Europe. The paper also indicates that this metric can help identify potential predictability drivers, with the example of a North Atlantic spring SST Atlantic preceding anomalously warm summers over Europe.

Overall the scientific question is simple and clear, and the results are quite convincing. Additionally, the paper is well written and enjoyable to read. This paper deserves to be published in WCD, after considering a few mostly minor revisions.

The introduction and discussion sections are particularly short. It is an asset for the introduction since it is sufficiently precise to properly state the scientific question addressed, and goes straight to the point. On the other hand, the results shown throughout the paper raise a number of questions that would deserve to be at least pinpointed in the discussion, if the authors want to keep the scope as it is. The main point to address concerns the section focusing on predictability source, and is detailed hereafter.

Finally some figures need to be be improved, as suggested below, to facilitate the reading and understanding.

As suggested by the reviewer, all figures were remade to facilitate reading and understanding. Figures 5 and 6 have been removed together with section 3.2 due to important criticism from both reviewers (see further comments below).

Specific comments

L.42: "Tmax may be seen as a seasonal integrator of heatwave activity".

No doubt that your approach is valid, but wouldn't it have been simpler to use the seasonal average of daily mean temperature, rather than daily maximum temperatures ? The departure of daily mean temperatures from climatology also permits to identify heatwaves, and it integrates the full diurnal cycle, which may be relevant since night-time temperature is also impactful during heatwaves.

This is a good point, we have changed our approach as suggested by the reviewer and now we use mean daily temperature instead of Tmax in all relevant analyses and figures. The results and conclusions are qualitatively the same.

L. 86-88 : This is not obvious from your figures. Considering this comment, we probably expect to see blue shades over NW Europe in fig 2.c but it is barely visible on your maps: the contrast between dark gray and blue is weak. I would suggest either to improve the maps (you may remove country borders, or pick a lighter gray shade for non significant values, or alternatively hatch/stipple where values are statistically significant.) or to provide additional support to your assertion.

The figure has been changed as suggested by the reviewer, now only statistically significant values are displayed. The differences in Europe were generally not significant, for this reason the sentence has been removed.

If fig. 5d shows the mean 1994 and 2003 JJA, one might wonder if the SST dipole in the Atlantic is present equivalently in both summers, or prevails in one of the two summers. Since spring 2003 does not stand out as a season with particularly high spring SST index (fig. 5c), this question is even more intriguing. You could provide in appendix or supplementary material separate maps for both summers and comment on the specific case of 2003 (weak dipole in spring, stronger in summer ?)

Figure 5 has been removed together with section 3.2 due to important criticism from both reviewers. New Figures 5 and 6 have been added to further display the use of the selection procedure based on SNR in specific regions of the Northern hemisphere. The sections have been changed accordingly.

L. 144-146: In my view, the subsampling of members based on the May index for re-forecast initialized in May makes no sense because the ensemble spread is presumably too small. I guess Mid-Atlantic SST anomalies are somewhat persistent, and it takes time for ensemble members to diverge from each other (and from SST initial conditions too). I think a discussion on the spread would be valuable here.

See answer to previous comment.

Overall, in section 3.2, the distinction between spring and summer SST patterns is a bit overlooked, and gives an impression of 'cherry picking'. Which one (spring or summer) is better correlated with warm summer anomalies over Europe ? In figure 5 you build your index on spring patterns, then you show summer patterns for 1994 and 2003, but the 2003 spring index does not stand out, and this point is not even commented on. Finally, in the subsampling procedure, you admit that it does not work when the subsampling is based on the May SST index. Besides the question of the relevance of subsampling too early after initialization (see point above), how could your findings help forecasters in real-time ? Would it be more advisable for them to systematically subsample their summer forecast ensemble, even if there is very little match with the SST pattern, or should they (also? instead?) evaluate a kind of distance metric or spatial correlation for each member to the SST dipole pattern, because maybe the SST pattern is not relevant every year.

The reviewer raises very good points and gives useful ideas, however due to important criticism from both reviewers, we have decided to focus the manuscript on the initial analysis (3.1) and leave out completely section 3.2 dealing with sources of predictability.

References:

The reference of the MeteoFrance system is the following

Batté L., L. Dorel, C. Ardilouze, and J.-F. Guérémy, 2017: Documentation of the METEO-FRANCE seasonal forecasting system 8. Météo-France, 36 pp., https://www.umr-cnrm.fr/IMG/pdf/system8-technical.pdf.

Changed

L.38 Since you have retrieved the forecast data from the C3S Climate datastore, I think you should cite it and acknowledge properly, as recommended here: https://confluence.ecmwf.int/display/CKB/How+to+acknowledge+and+cite+a+Climate+Data+Store+%28CDS%29+catalogue+entry+and+the+data+published+as+part+of+it

Changed

And here: https://cds.climate.copernicus.eu/cdsapp#!/dataset/seasonal-original-single-levels?tab=doc

Technical corrections

L.48 Typo: It should read "from ERSSTv5 (Huang et al., 2017) and ERA5, respectively."

Figure 1 : according to fig. 1c and 1f, a few grid points have SNR values exceeding 1. Thus, this should also show in the colorbar of fig 1b and 1e which looks bounded by 1

Figure changed.

Figure 5 : in the caption, the reference to subplot (e) is missing

Figure deleted

Figure 6: When zooming in, I can see gray dots everywhere. This figure should be improved.

In addition, the color bars are different for temperature and precipitation. The same color bar would help compare the skill improvements for both variables. Finally, the color bars themselves are confusing: they should be improved by aligning the labels with color bounds.

Figure deleted

L. 110-111: There is a mistake in the description of the ERA5 gray line in Fig5a/b, which gave me a hard time to understand: it is not the absolute value of the standardized ERA5 Tmax anomalies that you show, but the absolute deviation from mean! Luckily, the caption of figure 5 is okay.

Figure deleted along with section 3.2

L. 116: Missing specification that this statement refers to fig.5b

Figure deleted

L. 117-119 : For these years with high ensemble SNR and high absolute anomalies in observations, I assume that the sign of the model re-forecast anomaly is the same as the sign of the observed anomaly. If so, this should be stated in some way, or overlaid in fig 5a/5b if feasible.

Figure deleted

L. 142 : Fig 6a,c and 6b,d, not 5

Figure deleted

L.145: typo 're-forecasts'

Section removed

Review 2

We thank the anonymous reviewer for his/her useful and constructive comments

Review of "Exploiting the signal to noise ratio in multi-system predictions of summertime precipitation and maximum temperature in Europe" by Acosta Navarro & Toreti

The manuscript analyses the predictability of heat waves on seasonal scales and claims that by SNR analysis it is possible to improve seasonal prediction of temperature and precipitation. It hypothesises a physical mechanism for heat waves and checks its validity.

Generally there are a couple points which need clarification within the manuscript.

1. The abstract already states concerning SNR:

"Forecasts of maximum daily near surface air temperature and precipitation in boreal summers with high signal to noise ratio tend to coincide with observed larger deviations from the mean than years with small signal to noise ratio"

By the used definition of the SNR (l. 53) where the ensemble mean is divided by the standard deviation, this is an expected behaviour. Consequently, in section 3.1 (l.73 to l. 80) a sensitivity test is performed, including eliminating years of high or low SNR. When we follow this argumentation, then years with high SNR mean a large deviation of the mean and the other way round. As skill metric a correlation based metric (ACC) is used, which is well known that values at its extremes have a larger impact on the result than those close to its mean. In simple words: When a model is brave to predict a relative extreme event it is more likely to get a large penalty with correlation as if the model is relatively conservatively choosing the middle ground. Following from this I do not see as a reviewer what the scientific value of this analysis is apart from a statement that correlations as skill metrics are not perfect.

This is a fundamental problem within the main argumentation of the manuscript, which puts the overall results in doubt. A clarification would be required and how this really can be applicable to future predictions. For example the analysis of one specific grid point with demonstrating the described effects would certainly help in case the argumentation above is not valid.

We would like to stress that the absolute deviation from the mean is only one of the two factors that determine the value of the SNR, the other being the ensemble standard deviation. One cannot assume that large SNR are exclusively determined by a large deviation from the mean, but could also be caused by low standard deviations (i.e. high ensemble coherence) or a combination of both. But even under the assumption of constant standard deviation every year, extremes only have a larger impact on the correlation (ACC) if there is a statistical significance between the variables evaluated. If these values are independent then the correlation should be zero, regardless of whether extremes are considered or excluded in the sample, given a large sample. Furthermore, other metrics such as the mean square skill score or the Gilbert skill score (also known as equitable threat skill score) give qualitatively similar results. We have added the new Figures 5 and 6 focusing on several northern hemisphere mid-latitude regions.

2. Section 3.2 switches from a statistical analysis to a physical one and without prior introduction hypothesize about a physical mechanism of heat waves. That's said, there is a vast amount of literature about heat waves and this mechanism, which is simply ignored by the authors. Much more, the framing of section 3.2 let a reader assume that the authors came up with this mechanism on their own accounts. They use the discussion to name two citation, which already in part applied the mechanism and justify their results, but not as appropriate start with those and others in the manuscript and see how this can then be tested. This leads also to problematic definitions. E.g., they introduce without any justification and physical analysis a new physical mode within the North Atlantic (dipole) while the overall literature almost exclusively talks about a tripole. They acknowledge this in their discussion by citing Dunstone et al 2019 to justify their choice, but this should have been the other way round. Also an explanation of the physical mechanism is missing how over the North Atlantic the wave guide causes the observed dipole. This suggest an evaluation on the specific literature has not taken place and would be strongly suggested.

We acknowledge that this is a very important point raised by the reviewer. For this reason, we have decided to remove former Section 3.2 and focus and expand former Section 3.1 (See the reply to reviewer 1).

These two points are very serious and put the results and the validity/novelty of the presented results in doubt. I will recommend here a major revision, because I acknowledge that there could be misunderstandings on my side that do not justify a rejection right away. Anyway, a major revision would likely require a major rewriting of the manuscript as the current one asks potentially serious questions about the applied scientific methodology.

More detailed:

l. 50ff: A reference for the SNR equation would be a good addition.

Added

l. 64: "The patterns of SNR mirror those of ACC." This is not surprising, this is usually covered by a large amount of literature in the field under the name signal-to-noise paradox. It would be necessary to show by the authors, that this is really different than this established theory (Scaife & Smith 2018, https://doi.org/10.1038/s41612-018-0038-4 and references therein).

The studies related to the signal to noise paradox have been added and discussed in the introduction..

l. 100: In case the authors wish to show here a difference between Fig. 2 and 4, they have to plot the difference. It is not possible to derive a proper estimation of the difference from two absolute values alone. Furthermore, it is not unexpected to have massive differences between May and June, especially in Europe. Not only is there less of a lead time, but also the physical system changes from spring to summer.

Seems like there may be a misunderstanding. The forecast target months are always JJA, the only difference between them is that for Fig. 2 the forecasts are initialised in May, while in Fig. 4 the forecasts are initialised in June. The text has been rewritten to clarify this point. It

is quite common to have less skill with increasing lead time, especially if the first forecast month is considered, as it is the case of Fig. 4. During this first forecast month the impacts of initialization of the atmosphere are still very clear (hence considerably higher values). With the updated figures, the higher skill scores in Fig. 4 are clearly visible by naked eye.

l. 127 ff: This section requires a major revamp. When a physical mechanism is hypothesized, it would require a proper description of a physical mechanism on the basis of literature. Concerning the physical mechanism and its model representation there can be found plenty in the already cited Neddermann et al 2019. E.g. Wulff et al describe the mechanism in detail (without the side step of any dipole or tripole in the North Atlantic). The therein cited Ghosh et al 2017 gives further clarity on the connection between pressure fields and the waveguide. Concerning precipitation the duality of a waveguide induced East Atlantic Pattern and a summer NAO (Folland et al 2009) might be of interest. Dunstone et al 2018 (it is wrongly named 2019 in the manuscript) gives one indication. But the theory about tripole to Heat wave is older (Czaja & Frankignoul, 2002). More on background towards heat waves can also be found in the citation in the recent publications of Beobide-Arsuaga et al 2023 and Rousi et al. 2022. Nevertheless, with that said, it requires to proof that this manuscript brings additional value to the scientific discourse.

Please see the reply to the second major comment.

l. 129 ff: Reproducability is not given with this level of description of the index definition you use. Please use either an established index or include here an exact derivation of the one you chose.

This section has been removed (See response to major comments above).

Fig. 6: Significances are not clearly identifiable. Partly due to lack of the Figure resolution, but also the way it is marked. It makes it hard to distinguish between the significant and non-significant areas.

This figure has been removed (See response to major comments above).

---

## Referee Report (RR1)

Review of he revised manuscript  egusphere-2023-194 by Acosta-Navarro and Toreti

**General comments**

The authors chose to delete the section about the potential predictability drivers of dry and hot summers in Europe. Even if this reduces somewhat the scope and impacts of the study, it does improve the quality of the manuscript.
The comments have been adequately addressed by the authors. I only regret that the authors do not evaluate the contribution of the ensemble spread in the highest values of SNR, with respect to that of the deviation from the mean.
Overall I think the manuscript can now be published, if possible after addressing the minor comments indicated below.

Minor points:

Caption figure 5 : a word is missing "Skill metrics **are** provided separately"

Figure 5: A visualization of the 6 domains on a map would be appreciated. Either by adding a dedicated map, or by overlaying the 6 boundary rectangles on top of a map from figure 4.

Figures 5 et 6 : It would be worth assessing the statistical significance of the correlations in each focus region. This significance is important in your manuscript since you subsample your 24-year period into a reduced sample of years.

The year for the "Batté et al." reference is 2021, not 2017 (It was my mistake in the first round of review)

---

## Author Response (AR2)

Review 1

We kindly thank the anonymous reviewer for reviewing the manuscript a second time. We believe that the suggestions have helped improve the quality of the study.

General comments

The authors chose to delete the section about the potential predictability drivers of dry and hot summers in Europe. Even if this reduces somewhat the scope and impacts of the study, it does improve the quality of the manuscript.

The comments have been adequately addressed by the authors. I only regret that the authors do not evaluate the contribution of the ensemble spread in the highest values of SNR, with respect to that of the deviation from the mean.

We may have missed this comment in the first round. We have added a new figure (new Figure 2) and discussion, which aim at singling out from the SNR the impact of ensemble coherence on skill. We believe this new figure 2 brings an interesting perspective to the study and helps form a better understanding of how the two aspects of SNR (deviation from mean and ensemble coherence) relate to skill.

Overall I think the manuscript can now be published, if possible after addressing the minor comments indicated below.

Minor points:

Caption figure 5 : a word is missing "Skill metrics are provided separately"

Added

Figure 5: A visualisation of the 6 domains on a map would be appreciated. Either by adding a dedicated map, or by overlaying the 6 boundary rectangles on top of a map from figure 4.

Added

Figures 5 et 6 : It would be worth assessing the statistical significance of the correlations in each focus region. This significance is important in your manuscript since you subsample your 24-year period into a reduced sample of years.

Added the p-values of the linear correlation coefficients.

The year for the "Batté et al." reference is 2021, not 2017 (It was my mistake in the first round of review)

Changed

Review 2

We kindly thank the anonymous reviewer for reviewing the manuscript a second time. We believe that the suggestions have helped improve the quality of the study.

First of all I would like to thank the author to reduce the manuscript and focus on the statistical section. Whether it still fits the scope of the journal is a different question, which I am not able to answer.

I think that the manuscript has to be seen as the observation of a phenomena rather than its explanation. It lacks currently the theoretical argumentation for the described phenomena, but it has to be acknowledged, that the general foundations around the signal-to-noise paradox are shaky. Consequently, another observation contributing to the collection of observations around this phenomena might be acceptable at the moment. I currently cannot see any major flaws in the manuscript themselves, so an acceptance of it might be appropriate from that perspective. Nevertheless, as the authors created here a purely statistical paper, it would be wise for them to at least attempt a theoretical explanation to connect ACC with SNR. Therefore, small corrections at this point.

89: Here it is adviced to see whether a mathematical explanation for the connection between ACC and SNR can be found.

Although not directly a mathematical explanation, we have added a new figure (new Figure 2) and discussion, which in our opinion helps form a better understanding of how SNR relates to skill. We do so by separating the impact of the multi-system ensemble coherence on ACC. It clearly shows that in the tropics and sub-tropics the ensemble coherence is an important indicator of skill, especially for T2m. This result, in our view, may help shape future research dealing with the still obscure signal to noise phenomenon in climate science.

168: "Overall precipitation predictability is lower than T2m predictability in the regions analized, since skill scores for precipitation are generally lower than those of T2m."

That is the symptom, not the cause. Reason is of course the problem to predict complex dynamical, highly spatially variable variables like precipitation with an ESM. So reformulating it would be appropriate (so either with correct cause, or the statement, that you identify predictability by difference ein skill scores)

We consider this comment valid, therefore we have rephrased the sentence as suggested by the reviewer.